# Exploring the Numerical Reasoning Capabilities of Language Models: A Comprehensive Analysis on Tabular Data

**Mubashara Akhtar[1][*], Abhilash Shankarampeta[2][*], Vivek Gupta[3], Arpit Patil[4]**
**Oana Cocarascu[1] and Elena Simperl[1]**

[1]King's College London  [2]Meesho  [3]University of Pennsylvania  [4]University of Utah

mubashara.akhtar@kcl.ac.uk
abhilash.shankarampeta@meesho.com

## Abstract

Numbers are crucial for various real-world domains such as finance, economics, and science. Thus, understanding and reasoning with numbers are essential skills for language models to solve different tasks. While different numerical benchmarks have been introduced in recent years, they are limited to specific numerical aspects mostly. In this paper, we propose a hierarchical taxonomy for numerical reasoning skills with more than ten reasoning types across four levels: representation, number sense, manipulation, and complex reasoning. We conduct a comprehensive evaluation of state-of-the-art models to identify reasoning challenges specific to them. Henceforth, we develop a diverse set of numerical probes employing a semi-automated approach. We focus on the tabular Natural Language Inference (TNLI) task as a case study and measure models' performance shifts. Our results show that no model consistently excels across all numerical reasoning types. Among the probed models, FlanT5 (few-/zero-shot) and GPT-3.5 (few-shot) demonstrate strong overall numerical reasoning skills compared to other models. Label-flipping probes indicate that models often exploit dataset artifacts to predict the correct labels.[1]

## 1 Introduction

Numerical data is ubiquitous in the real-world. Many applications in domains such as finance, economics and science require understanding and reasoning with numbers. In recent years, benchmarks were introduced to study language models' numeracy skills (Zhang et al., 2020; Wallace et al., 2019; Dua et al., 2019). However, these datasets mostly concentrate on few, specific numerical reasoning types (e.g. scales (Zhang et al., 2020)).

---

[*]Equal contributions
[1]Data and code are available at https://github.com/mubasharaak/numerical_reasoning.

| Hulk | |
|---|---|
| **Directed by** | Ang Lee |
| **Release date** | June 20, 2003 |
| **Running time** | 138 minutes |
| **Budget** | $137 million |
| **Box office** | $245.4 million |

**H1:** Hulk was released on 20th June, 2003. *(E)*
**Date:** Hulk was released on 20-06-2003. *(E)*
**Date Flip:** Hulk was released on 12-08-2009. *(C)*

**H2:** The movie has a length of 138 minutes. *(E)*
**Appr:** The movie has a length of about 150 minutes. *(C)*

**H3:** The movie can be watched in about two hours. *(E)*
**Num:** The movie can be watched in about 2 hours. *(E)*
**Num Flip:** The movie can be watched in 1 hours. *(C)*

**Arith:** Hulk brought in $108.4 million profit. *(E)*
**Arith Flip:** Hulk brought in $120.9 million profit. *(C)*

Table 1: Base hypotheses (H1, H2, H3) and (**flip**ped) probes for heterogeneous numbers (i.e. **date**), **appr**oximation, **num**eracy, and **arith**metic. Labelled as **E**ntail or **C**ontradict.

Moreover, evaluating models on numerical benchmarks, it often remains unclear why models struggle with the tasks. For example, the issues can arise from models struggling to recognize numerical representations in text, failing to compute arithmetic operations, or predicting incorrect outputs due to a lack of numerical commonsense knowledge. We aim to explore these questions in greater detail in this study. Limitations of language models' numerical abilities, as discussed in prior research, include tokenization and representation of numbers in text (Thawani et al., 2021b), hallucination (Ji et al., 2023; Chen et al., 2023; Ye et al., 2023), and generalizability/robustness issues (Razeghi et al., 2022; Geva et al., 2020; Xu et al., 2022).

Successful numerical reasoning requires a combination of skillsets: understanding representation of numbers (Thawani et al., 2021a,b) and their meaning in a given context (Loukas et al., 2022), applying operations (Geva et al., 2020; Patel et al., 2021), and integrating factual and commonsense numerical knowledge to solve numerical

problems (Lin et al., 2020; Park et al., 2022). For example, classifying the hypotheses *"The movie can be watched in about 2 (or 'two') hours."* from Table 1 requires understanding that both *"2"* and *"two"* depict the same numerical value, converting *"2 hours"* to another unit (i.e. 120 minutes), and applying approximation to map *"120 minutes"* to *"138 minutes"* in the table.

In this paper, we evaluate state-of-the-art models on various numerical reasoning types. To assess which reasoning types are challenging for specific models, we create a diverse and large set of numerical probes and measure shifts in models' performance. We organize all probed reasoning types in a hierarchical taxonomy. Inspired by how humans understand and reason with numbers, as well as previous numerical benchmarks, we include eleven reasoning types across four level: *representation*, *number sense*, *manipulation*, and *complex reasoning* (Figure 1). We apply a semi-automated approaches for probe creation. We select tabular NLI (TNLI) as a case study task, given three criteria: $(i)$ numerical data (numbers, percentages, dates, etc.) is prevalent in tables; $(ii)$ tables are common in real-world data sources such as in scientific publications, database systems and financial documents; $(iii)$ tables as structured data facilitate automated perturbations to create large-scale probing sets. See Table 1 for some examples of probes created from hypotheses (H1, H2, H3) and the given table.

Our experiments conclude that large language models (LLMs) like FlanT5 and GPT3.5 perform better than other models on various numerical reasoning tasks. Both table-based and numerical models struggled to understand data with flipped labels and negative values. Moreover, we observe that some models' performance improves significantly for counterfactual probes (e.g. NT5 and TAPAS) and label-flipping probes (e.g. FlanT5 zero-shot), which indicates that models might exploit dataset artifacts and are biased towards one label. These findings emphasize the importance of further systematically investigating numerical reasoning capabilities across various NLP models. Our **contributions** are as follows:

- We introduce a taxonomy for numerical reasoning skills, including representation/number sense/manipulation skills and complex reasoning with numbers.

- We propose a semi-automated approach to

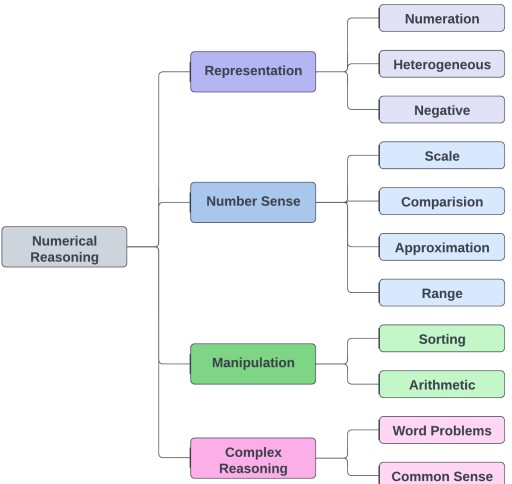

Figure 1: Overview of numerical reasoning types.

create large-scale, numerical probe sets using table NLI datasets.

- We evaluate three different categories of language models (LMs) on our numerical probe sets: $(i)$ numerical LMs; $(ii)$ LMs for tabular data; and $(iii)$ zero-/few-shot LMs.

## 2 A Taxonomy for Numerical Reasoning

This section introduces a hierarchical taxonomy for numerical reasoning, inspired by previous works on numeracy in NLP (Thawani et al., 2021b; Xu et al., 2022) and psychology (Barrouillet and Fayol, 1998a; Whyte and Bull, 2008; Bofferding, 2019). We group numerical reasoning skills given their complexity level in four categories: $R1 - R4$.

### 2.1 Number Representation ($R1$)

This category includes skills for understanding the *form* of numerical data. Similar to the notion of form in language (Bender and Koller, 2020), this is the realization of numbers in text; the way they are represented and expressed.

**Numeration.** Numeration studies language model's understanding of representation systems common for numbers in English: the Arabic ("2") and English ("two") numeration systems. Specifically, we probe if LMs can link between distinct symbols used for the same number. For example in Figure 1, H3 contains "two" as a word, which can be also represented through "2".

**Heterogeneous Number Types.** Formatted numbers (e.g. dates, times, and fractions) are frequently used to convey additional information associated with a numerical value. Numbers are formatted in a specific way given their context and purpose,

such as expressing times and dates using full-stop ("."), using the "%" symbol to indicate fractions, and different currency symbols for money (i.e. "$" or "€"), e.g. H1 and "Arith" in Figure 1.

**Negative Numbers.** Early on in their development, children develop some mental model for negative numbers (see experiments with first-graders in Bofferding (2019)). Using negative numbers requires understanding the notation of negatives (i.e. "−" followed by a number). This also includes distinguishing between minus in subtractions ($1 − 3$), dates (12-12-2022), counts (i.e. from one to three) and in negative number ($−2$).

## 2.2 Number Sense ($R2$)

Number sense includes reasoning skills for conceptualizing number quantities and understanding their meaning in a given context.

**Scale.** In everyday communication, numbers commonly occur with measurement scales, e.g. weights, distances, or heights. Understanding numbers in context of scales is a basis for various applications, e.g. question answering (e.g. *"We are driving* $80$ *km/h, is this within the speed limit?"*), commonsense (e.g. *"Cats weight between four and five kilograms."*) (Lin et al., 2020), and temporal reasoning (e.g. *"She left the office thirty minutes ago."*) (Zhou et al., 2020; Zhang et al., 2020).

**Comparison.** Comparing numbers allows understanding numerical relationships. It involves identifying which numbers are greater than, less than, or equal to others. For example, given the table in Figure 1, understanding *"The running time of Hulk is longer than* $120$ *minutes."* requires comparison.

**Range.** The question *"Was the budget of the movie between* $130$ *and* $245.4?"* about the table in Figure 1 requires understanding number ranges. Already at an age between two and three years, children develop numerical abilities to understand sequences of numbers and start reciting numbers in an appropriate order (Fuson, 2012; Laski and Siegler, 2007). Models' that understand the notation of ranges, can correctly answer the question by knowing that 137 is in the range $130 − 245.4$.

**Approximation.** Humans commonly approximate number values in everyday life (Odic and Starr, 2018; Bonny and Lourenco, 2013). H3 in Figure 1 requires approximation among other skills to map "about two hours" to "138 minutes" in the table. As a reasoning skill, it allows to make quick estimations and metric unit conversations, and understand the approximate values of numbers without calculating them explicitly.

## 2.3 Manipulation ($R3$)

Manipulation reasoning types are used to apply basic operations on numbers such as addition. Successful manipulation of numbers requires understanding their *representations* and meaning in the given context (i.e. *number sense*).

**Sorting.** The sentence *"Out of all Ang Lee's directed movies, 'Hulk' was the one with the second highest box office income."* requires sorting all movies according to their box office income in order to select the one with the second highest income. Sorting objects according to some criteria is a basic milestone for developing cognitive skills. By age two, children already begin to understand the concept of sorting.

**Simple arithmetic.** Arithmetic reasoning is the ability of manipulating numbers with basic operations (addition, subtraction, multiplication, division). While adults commonly retrieve results of simple calculations from memory, children use different operations (Barrouillet and Fayol, 1998b).

## 2.4 Complex Reasoning ($R4$)

This category builds on all previous reasoning categories ($R1 − R3$) to solve numerical word problems (NWP). NWP are expressed through natural language and require multistep reasoning. Extracting information from the problem description and applying numerical/mathematical reasoning using the retrieved information and world/commonsense knowledge is required (Upadhyay and Chang, 2017; Amini et al., 2019; Huang et al., 2016).

## 3 Numerical Probing Framework

This section provides an overview of the probing framework. We use tabular Natural Language Inference (TNLI) for automated probe creation.

### 3.1 Preliminaries

**Tables for numerical probing.** Tables align well with our objectives given three key criteria: ($i$) numerical data is common in tables; ($ii$) tables are frequent in real-world data sources; ($iii$) tables, due to their structured formats, facilitate automated perturbations for probe creation. Tables'

semi-structured format, the alignments available between table cells and column/row headers, and the frequency of numbers, make them well suitable for creating numerical probes automatically.

**Table NLI.** Given a natural language sentence as hypothesis and a tabular premise, the aim of TNLI is to classify if the hypothesis *entails* or *contradicts* the table (Gupta et al., 2020). We use the table NLI datasets *TabFact* (Chen et al., 2020) and InfoTabs (Gupta et al., 2020), as well as recast the table QA datasets TAT-QA (Zhu et al., 2021) and TabMWP (Lu et al., 2023) to NLI (i.e. *TATQA-NLI*, *TabMWP-NLI*). TAT-QA includes metadata, i.e. annotations of cells and operations per correct answer. This information is not available for any TNLI dataset and is crucial to create probes for specific reasoning types, e.g. *arithmetic reasoning*. Table 2 provides an overview of the TNLI datasets.

**Preprocessing.** For each of numerical reasoning type, we first identify base TNLI hypotheses and/or tables in the datasets that can be used for automated probe creation. Hereby, we defined a list of reference tokens specific for each reasoning type and to filter relevant dataset samples. For example, we used units of measurements such as "hour", "meter", or "kilogram" filter hypotheses for *scale* probes (see §4 for more details). To recast the TAT-QA dataset, we follow the simple yet effective, rule-based approach proposed by Demszky et al. (2018) for QA to NLI conversion.

### 3.2 Probes through Structural Perturbation

Overall, we our framework includes three types of probes, created through hypotheses perturbation and counterfactual tables.

**1. Hypothesis label-preserving probes** We create label-preserving probes changing the base hypothesis such that its meaning is not changed and the initial label is preserved. They are used to evaluate model's ability to reason and predict the correct label given semantically-equivalent changes.

**2. Hypothesis label-flipping probes** To generate label-flipping probes, we modify the base hypothesis such that its meaning alters and the label of the probe flips, e.g. from entailment to contradiction. We aim to overcome potential dataset artefacts that might be exploited for label prediction instead of performing numerical reasoning.

These changes are specific to the reasoning types. For example, to flip labels for *scale* probes, we

| Dataset | Hypotheses | Tables | Num cells | Probes |
|---------|-----------|--------|-----------|--------|
| TabFact | 118,275 | 16,573 | 59.00% | 214,440 |
| InfoTabs | 23,738 | 2,540 | 53.6% | 19,779 |
| TATQA-NLI | 4,947 | 2,156 | 59.7% | 15,139 |
| ToTTo | 1,000 | 892 | 45.7% | 1,000 |
| TabMWP | 283 | 283 | 38.3% | 238 |

Table 2: TNLI probing datasets; *num cells* refers to the average ratio of numerical cells in tables.

substitute measurement units for a particular scale (e.g. "kilograms") by another unit (e.g. "meters") or introduce errors in conversion of units (e.g. 3 kilometers replaced by 3, 000 meters).

**3. Table Probes through Counterfactual Table Editing** We also probe with counterfactual tables to evaluate if models rely on spurious patterns in the premise table for label prediction. We filter the counterfactual datasets by Jena et al. (2022) consisting of *{hypothesis; original table; counterfactual table}* for numerical hypotheses.

## 4 Probing with TNLI Datasets

This section discussed probes in detail and how we created them for each reasoning type from §3.[2]

**Numeration.** To study models' understanding of string ("two") and numerical (e.g. "2") number representations, we create two types of numeration probes. One converting number representations from strings to numeric, while the second category applies the conversion vice versa. We filter hypotheses with numbers written as strings ("two") and substitute them by their numeric counterpart (e.g. "2"). The label-preserving probes are semantically equivalent to the base hypotheses and the label (e.g. *entailment*) is not changed. Label-flipping probes replace the converted number $x$ by a random number in the range of $[x - x * 0.5; x + x * 0.5]$. For example, the numeration flipping probe of H1 (Table 3) replaces 112 by one hundred and forty-four and flips the label from *entailment* to *contradiction*.

**Heterogeneous number types.** We created heterogeneous probes for the following categories frequent in the TNLI datasets: date formats, ordinals, percentage, currencies, and scientific notation. To filter base hypotheses, we applied a simple, rule-based approach specific to each category (i.e. dates formats, percentage, ordinals, etc.). To create label-preserving probes we applied representation-level changes which did not change the semantic meaning. For H3, we substituted 3rd June, 1986 by another English date format 03-06-1986. To flip the

---

[2]Find details on probe statistics in Appendix B.

| Rafael Nadal | |
|---|---|
| **Plays** | Left-handed |
| **Born** | 3 June 1986 (age 37) |
| **Height** | 1.85 m |
| **Turned pro** | 2001 |
| **Prize money** | US$116,111,561 (3rd all-time leader in earnings) |

| | |
|---|---|
| *Base Hypothesis $H_1$* | Born in 1986 , Nadal is age 37 currently. |
| *Numeration Probe $H_1$* | Born in nineteen eighty six, Nadal is age thirty seven currently. |
| *Num Flip Probe $H_1$* | Born in nineteen ninety two, Nadal is age forty one currently. |
| *Range Probe $H_1$* | Born in 1986, Nadal is age between 31-43 currently. |
| | |
| *Base Hypothesis $H_2$* | The player's birth date is on 3rd June, 1986 . |
| *Heterog Probe $H_2$* | The player's birth date is on 03-06-1986. |
| *Heterog Flip Probe $H_2$* | The player's birth date is on 15-01-1999. |
| | |
| *Base Hypothesis $H_3$* | With $116,111,561 prize money, he is the 3rd highest earning all-time player. |
| *Heterog Probe $H_3$* | With $116.111561e - 6$ prize money, he is the third highest earning all-time player. |
| *Approx Probe $H_3$* | With about $116, 000, 000 prize money, he is the 3rd highest earning all-time player. |
| | |
| *Base Hypothesis $H_4$* | Rafael Nadal has a height of 1.85 meters. |
| *Scale Probe $H_4$* | Rafael Nadal has a height of 185 centimeters. |
| *Scale Flip Probe $H_4$* | Rafael Nadal has a height of 5.2 ft. |
| | |
| *Base Hypothesis $H_5$* | After the year 2000, the player Nadal turned pro. |
| *Comparison Probe $H_5$* | After the year 1995, the player Nadal turned pro. |
| *Comparison Flip Probe $H_5$* | Before the year 1990, the player Nadal turned pro. |

Table 3: Exemplary hypotheses and non-/flipping probes for evaluated reasoning types

label, we replaced the date in the adjusted format by a random date, i.e. 15-01-1999. We replaced percentage signs by the token *"percentages"* and vice versa. Similarly, ordinals written as words (*first*) were exchanged by numerical representations ($1st$) and the other way around. For hypotheses with large numbers (e.g. "$116,111,561" in H3), we introduced scientific notations ($116.111561e - 6$).

**Negative numbers.** To create negative probes, we replaced negative numbers $-n$ (e.g. $-3$) by string equivalents (e.g. *minus* 3; *negative* 3) and evaluated changes in model performances on these semantically same sentence pairs. For label-flipping probes, we converted negative numbers into the positive counterpart $n$. For example, converting *"The company's monthly closing resulted in -5 million USD."* to *"The company's monthly closing resulted in 5 million USD."* flips the label.

**Scale.** We created two types of scale probes: ($i$) *conversion*; ($ii$) *mapping*. Conversion convert numbers within a measurement scale. For $H4$ in Table 3, we converted the number and measurement unit ( 1.85 meters ) to the next smaller unit within the same scale (185 centimeters) for the label-preserving probe. For label-flip, we introduced an error in the converted number, i.e. converting 1.85 meters. to 5.2 ft instead of 6.07 ft.

Mapping probes replace the number and measurement unit by an equivalent (e.g. $1.85m$ by 1.85 meters) for label-preserving probes and a random measurement unit e.g. $1.85m$ to 1.85 kilograms) to flip the base hypotheses.

**Comparison.** We first created a list of signal word-pairs by prompting GPT3.5. The list includes pairs such as {"bigger":"smaller"}, {"taller":"shorter"}, and {"faster":"slower"}. Using these pairs and their synonyms, we filtered base hypotheses and created three types of comparison probes. First, changing the signal word with its opposite counterpart to flip labels (see H5 in Table 3 flipping "after" to "before"). Second, altering the number such that the comparison and label do not change: replacing "after 2000" by "after 1995" ($H5$). Finally, we combine both prior approaches to create label-flipping probes, e.g. "Before the year 1990, the player Nadal turned pro."s

**Approximation.** We first extract a number $n$ from our base hypothesis and given the value of $n$, we decide the magnitude of rounding to apply. While smaller numbers are rounded to tens, larger number are rounded to hundreds, thousands or larger decimal points. For example, we created the probe *"With about $116, 000, 000 prize money, he is the 3rd highest earning all-time player"*

by rounding $n$ equal $116,111,561 to "about $116,000,000" (H3 in Table 3).

**Range.** To create range probes, we substitute number $n$ in the base hypothesis by an appropriate range, e.g. 37 by "between 31-43" (H1). We define the radius of the range and its boundaries automatically given the value of $n$. For example, given $n < 10$, we randomly sample a radius between $1 - 5$. For $n = 7$ and a sampled radius of 2, the range will be $[5 - 9]$. We select decimal boundaries if $n$ is a decimal number.

**Sorting.** We utilized table columns as number sequences to create sorting probes. We generated a list of position indicators in number sequences (e.g. "top", "second" "3rd", "biggest", "lowest"). These words were used to filter base hypotheses. To create label-flipping probes, we changed the position of the sequence to another one. For instance, we modified "in the first quarter of 2018" to "in the third quarter of 2018" by selecting the value from the third row instead of the first.

**Simple arithmetic.** Using on TATQA-NLI its metadata indicating the involved numbers and operations for numerical reasoning, we created arithmetic probes. We extracted probes involving addition, subtraction, multiplication, and division. Additionally, we generated label-flipping probes by replacing the operation output (e.g. result of subtraction) in the hypothesis with a different number. In Table 1, the *"Arith"* probe involves calculating the difference between the *budget* and *box office* values to determine the correctness of 108.4. The flipped arithmetic probe produces a close but incorrect subtraction output, 120.9.

**Numerical word problems.** We converted TabMWP questions and answers into declarative hypotheses. TabMWP is a dataset of free-text math word problems that involve reasoning with tabular data. For label-flipping probes, we substituted numbers in the hypotheses with random numbers from the same column.

**Counterfactual Table NLI Probes.** We filtered the counterfactual ToTTo (Parikh et al., 2020) dataset by Jena et al. (2022) for numerical hypothesis. To create counterfactual tables, they swap two or more table cells to modify the tables such that the label of the respective hypothesis changes from entailment to contradiction and vice versa.

## 5 Experiments and Analysis

Next, we provide an overview of all models that were evaluated on the probes from §4. We also discuss the obtained results and insights.

### 5.1 Probed Models

We use state-of-the-art models which are divers in terms of architecture, size, and training setup, grouped into three categories:

($C1$) **Numerical LMs.** This category includes LMs adapted for numerical reasoning. *LUNA* (Han et al., 2022) is a recent transformer-based model with an adapted tokenization approach for numbers. The model encodes numbers as single tokens (e.g. $3, 201$) instead of splitting them down into subwords or binned tokens. *NT5* (Yang et al., 2021) is a variation of the T5 model. It has been modified for numerical reasoning through additional pretraining objectives and fine-tuning using numerical datasets. *PASTA* (Gu et al., 2022) is based on DeBERTa and is pretrained with objectives that use table-based numeric operations.

($C2$) **LMs for tabular reasoning.** *TAPAS* (Herzig et al., 2020) extends the BERT encoder with table-specific embeddings. We used a TAPAS model trained with intermediate pretraining on synthetic and counterfactual data (Eisenschlos et al., 2020). We also probe TAPEX (Liu et al., 2022), which uses BART (Lewis et al., 2020) as its base model and pretrains the model to mimic a neural SQL executor over tables. Similarly, ReasTAP (Zhao et al., 2022) is a BART-based model pretrained on synthetically generated data requiring seven table reasoning skills, including a numerical task, temporal reasoning, and conjunction. Previous works have also shown the success of the *\*BERT* models on tabular NLI tasks (Herzig et al., 2020; Yin et al., 2020; Shankarampeta et al., 2022; Akhtar et al., 2022). Tables are either linearized or processed into sentences or structured formats. The transformed tables are then used as input to the models. We used a DeBERTa model (He et al., 2021) trained on multiple NLI datasets for this setting.

($C3$) **Large LMs.** For few-/zero-shot evaluation, we selected FlanT5 (Shen et al., 2023), GPT3.5, and PaLM 2 (Chowdhery et al., 2022) and probed them in both a few-shot and zero-shot setting.

| Model | Table Specific | | | Numerical Specific | | | | Large LMs | | | | | |
|---|---|---|---|---|---|---|---|---|---|---|---|---|---|
| | | | | | | | | FlanT5 | | GPT3.5 | | PaLM | |
| Reasoning | TAPAS | DeBERTa | TAPEX | NT5 | LUNA | PASTA | ReasTAP | few | zero | few | zero | few | zero |
| **Representation** | | | | | | | | | | | | | |
| Numeration | -0.32 | -1.82 | -7.84 | -4.18 | -5.22 | -7.7 | -7.18 | 1.28 | -8.84 | -0.47 | 5.65 | -1.35 | -3.07 |
| Heterogeneous | -4.03 | -2.36 | -5.94 | -3 | -10.09 | -7.76 | -3.18 | 0.34 | -5.49 | 6.8 | 6.65 | 0.44 | -2.22 |
| Negative | -46.11 | -13.77 | 0.56 | -94.48 | -75.55 | -10.68 | 2.65 | 19.21 | 42.3 | 8.24 | 2.3 | -2.17 | 1.14 |
| **Label Flipped** | | | | | | | | | | | | | |
| Numeration | -38.87 | 4.09 | -43.3 | -48.53 | -71.35 | -25.85 | -37.21 | -78.37 | 33.38 | -37.78 | 44.71 | -37.29 | -46.45 |
| Heterogeneous | -9.57 | 8.53 | -32.25 | -1.97 | -43.48 | -23.59 | -16.21 | -53.44 | 86.6 | -27.97 | 27.79 | -20.65 | -25.31 |
| Negative | -64.81 | -41.56 | -97.01 | -17.87 | 76.85 | -70.58 | -96.46 | -83.92 | 173.14 | -63.64 | 2.2 | -80.43 | -78.41 |
| **Number Sense** | | | | | | | | | | | | | |
| Scale | 0.03 | -6.25 | -12.91 | 1.21 | -11.43 | -1.56 | -4.6 | -9.45 | -7.05 | 2.46 | -0.52 | -3.71 | -17.58 |
| Comparison | -21.8 | -18.18 | -12.58 | -29.19 | -30 | -35.11 | -40.02 | 29.38 | 140.82 | -9.39 | 9.13 | -16.91 | -20.08 |
| Approximation | -5.61 | -6.65 | -18.9 | -9.55 | -7.67 | -27.44 | -7.81 | -9.66 | -12.94 | 0.03 | 12.08 | -10.44 | -12.03 |
| Range | -18.89 | -33.77 | -1.96 | -20.43 | -86.77 | -84.66 | 4.97 | 22.44 | 178.13 | 0.5 | -1.07 | 14.37 | 4.41 |
| **Label Flipped** | | | | | | | | | | | | | |
| Scale | -23.73 | -64.58 | -30.41 | -39.08 | -68.44 | -51.66 | -16.54 | -69.56 | 93.77 | -39.08 | 39.98 | -17.85 | -27.4 |
| Comparison | 57.67 | -19.36 | -4.83 | -29.62 | -0.28 | -19.1 | -15.65 | -8.47 | -40.75 | -20.81 | 14.67 | -17.96 | -16.09 |
| **Manipulation** | | | | | | | | | | | | | |
| Sorting | -34.8 | 28.66 | -91 | -22.6 | 54.31 | -4.9 | -83.96 | -86.67 | 25 | -57.39 | -5.62 | -32.45 | -39.59 |
| Arithmetic | -58.62 | -24.96 | -95.53 | -27.1 | 7.07 | -49.06 | -88.87 | -71.53 | 265.07 | -60.34 | 4.87 | -67.67 | -64.98 |
| **Complex Reasoning** | | | | | | | | | | | | | |
| Complex | 63.37 | 6.93 | -80.18 | -3.22 | 41.41 | -50.84 | 116.17 | -89.77 | -40 | -60.22 | -4.35 | -73.4 | -75.9 |
| Counterfactual | 44.5 | 55.54 | -12.29 | 159.3 | 0.98 | -6.09 | -10.12 | 61.5 | 12.23 | 40.63 | 5.26 | 30.57 | 48.56 |

Table 4: Probing results given as accuracy difference (in %) between base hypotheses and probes.

## 5.2 Training and Evaluation

To fine-tune models, we used the base hypotheses of the training datasets (e.g. InfoTabs) and evaluated models only on probes created with their testsets. The few-shot models were prompted with 2-shot extrapolation. We evaluated all models in a 3-step process: (1) evaluation of base hypotheses $H$; (2) evaluation of probes $P$, created using $H$; (3) calculating changes in model performance by comparing accuracy of $P$ to $H$. As our TNLI task is a binary classification task, we used accuracy for evaluation.

## 5.3 Results and Discussion

Table 4 gives on overview of all probing results. If available, we separately list scores for flipped probes, e.g. *numeration* and *numeration flipped*.

($Q1$) **Does any model excel in all numerical reasoning types?** While there is not one best-performing model across all reasoning types and different models struggle with different types, FlanT5 and GPT3.5 show overall good performance in a zero-shot setting. While GPT3.5 (few-shot) performance drops by $-60.22\%$ for complex reasoning probes, the model's average accuracy change is around $-16.7\%$ for other types. This can be related to (1) models pretraining data, and (2) training on chain-of-thought reasoning tasks (Wei et al., 2022). GPT3.5 was trained on more than 300 TB data Common Crawl, allowing the model to memorize much more numerical data than other probed models. In comparison, DeBERTa was

trained on only $78GB$ of data (He et al., 2021). Interestingly, both NT5 and FlanT5 use T5 as their base model. FlanT5 was instruction-fine-tuned and outperforms NT5 many probing categories.

($Q2$) **What factors can contribute to high performance variations across certain reasoning types?** Large performance variations mainly occur due to inconsistent numerical reasoning of models across tasks. For example, we observe that some models struggle with more basic reasoning (e.g., FlanT5 zero on numeration) while performing very well on more complex types. This behavior might have different reasons. One potential reason is memorization. Previous works (Petroni et al., 2019; Carlini et al., 2021; Ishihara, 2023) show that large pretrained language models store knowledge in their parameters, which they tend to retrieve instead of reasoning over the provided input (Gupta et al., 2022a). Hence, models can memorize common arithmetic operations they encounter during training and perform well on certain downstream tasks. For example, flipping numbers as words ("two") to numerals ("2") might allow models to retrieve knowledge which they didn't consider for the initial hypothesis. Another reason for high-performance drops can be the hallucination of models. While models initially perform well on hypotheses, adjusting the numbers can hinder models from relying on spurious patterns.

($Q3$) **How do models perform on different types of numerical reasoning? Representation.** In

Table 4, comparing numeration probes, we find for all models a performance drop of between $[0; -10]$ percentages for numeration probes, except FlanT5 (few). This drop strongly increases for almost all models evaluated numeration flipped probes. For example, FlanT5 (few) shows a performance drop of $-78.37\%$. FlanT5 (few) also performs well on heterogeneous probes, followed by DeBERTa ($-2.4\%$) and NT5 ($-3\%$). Whereas LUNA performance drops significantly for heterogeneous probes (flipped and non-flipped). TAPAS, NT5, and LUNA show significant performance drops (between $-38.87\%$ and $-71.35\%$) on negative number probes. This could be because the models exploit correlations between the "$-$" sign and labels for predicting base hypotheses. Interestingly, few- and zero-shot models like FlanT5 and GPT3.5 show improvements on negative number probes. This may be because the models understand probed versions of negative numbers (e.g. "minus 22") as a negative number but not the initial representation (e.g. "$-22$").

**Number sense.** Comparing models based on number sense probes, we observe different patterns for fine-tuned models and few-/zero-shot models. Fine-tuned models struggle especially with comparison probes, with a $-26.7\%$ average performance drop. Scale probes show a $-42.1\%$ decrease on flipping probes, while approximation (flipping) probes report a $-12.0\%$ decrease in model performance. In contrast, FlanT5 perform better on comparison and range probes, sometimes surpassing predictions on the base hypotheses. All models demonstrate lower performance on approximation probes compared to the base hypotheses, with PASTA performance dropping by $-27.44\%$.

**Manipulation and Complex Reasoning.** Fine-tuned models exhibit an average accuracy drop of $-57\%$ on arithmetic probes, except for LUNA with a slight performance increase. The performance of PaLM (zero) and FlanT5 (few) drops by $-67.67\%$ and $-71.53\%$, respectively. All models' performance drops on sorting probes (avg. $-27\%$), except for DeBERTa, LUNA, and FlanT5 (zero). Unlike most other reasoning types, fine-tuned models outperform few-/zero-shot models on complex reasoning probes. ReasTAP achieves the highest accuracy, followed by TAPAS and LUNA. FlanT5, TAPEX, and PaLM have the largest performance drops on complex reasoning probes.

($Q4$) **Do models perform similarly for flipped and non-flipped probes?** We observe higher performance drops for label-flipping probes compared to non-flipping probes across models. Models that struggle with flipping probes but perform well on their non-flipping counterparts indicate a reliance on spurious patterns for label prediction. The performance of TAPAS, TAPEX, PASTA, ReasTAP, FlanT5 (few), and PaLM drops significantly for the representation reasoning category comparing non-flipping and flipping probes. For example, TAPAS performance drops by $-2.28\%$ on numeration probes, but show a drop of $-45.98\%$ on numeration flipping probes. Similarly, DeBERTa performs well on scale probes ($-6.25\%$) compared to the flipping version ($-64.58\%$). PaLM performance on numeration, heterogeneous, and negative probes drops by approximately $-35\%$, $-20\%$, and $-80\%$ on flipping counterparts. DeBERTa exhibits robust performance on number flipping probes for sorting and FlanT5 on negative numbers, as well as arithmetic probes.

($Q5$) **Are numerical and table-specific models better for numerical reasoning than LLMs? Numerical models.** Our experiments do not indicate any superiority of numerical models over others. LUNA, a transformer model that uses a specific tokenization method for numbers, performs similarly to other models on many reasoning types. The only reasoning type where LUNA outperforms is comparison flipping probes, with a small improvement of $0.28\%$. PASTA is a DeBERTa-based model trained on numerical data and pretraining objectives. However, compared to DeBERTa, it only performs better on negative number and scale probes.

# 6 Related Work

**Numeracy Taxonomies in NLP.** Prior works have introduced surveys and taxonomies to organise numeracy in NLP research. Thawani et al. (2021b) categorise NLP work on numeracy into seven subtasks along the dimensions granularity (i.e. exact and approximate numbers) and unit (i.e. abstract and grounded numbers). Xu et al. (2022) focus on the robustness of QA systems in handling numerical data and organize their numeracy probing tasks in two broad categories: ($i$) numerical parsing, and ($ii$) semantic parsing. The DROP benchmark (Dua et al., 2019) is a QA dataset that requires discrete operations (e.g. subtraction, count,

sort) to answer questions over text. While Thawani et al. (2021b) concentrate on number representations in NLP systems, our work includes three further numerical reasoning categories. Xu et al. (2022) focus on the robustness of NLP models in handling numerical data. Our probing study on the other side pivots towards the reasoning capabilities of models when dealing with numerical and tabular data. Different to prior work, our study gives a broad and in-depth evaluation of ten different models from three different categories (numerical, tabular, large pretrained LMs) on more than ten different reasoning types (representation, number sense, manipulation, and complex reasoning).

**Language Model / Numerical Skills.** Various studies have evaluated LMs' numerical skills in recent years. Earlier works probed word embeddings for numeration (e.g. 4=four) (Naik et al., 2019), comparison (e.g. $3 < 4$) (Wallace et al., 2019), scale (Zhang et al., 2020), and superlatives (Wallace et al., 2019). More recent works evaluate LMs on out-of-distribution numbers (Kim et al., 2021), numeration/magnitude/sorting/superlatives (Pal and Baral, 2021), and arithmetic (Muffo et al., 2022). Our work builds upon these previous evaluation studies and extends them with further numerical probing categories, e.g. heterogeneous numbers.

**Numerically-tuned Language Models.** Various numerical LMs have been developed in recent times. Geva et al. (2020) and Liang et al. (2022) inject numerical skills into BERT through numerical pretraining objectives. PASTA (Gu et al., 2022) and NT5 (Yang et al., 2021), which are based on DeBERTa and T5 respectively, fall into the same category of models. Another line of work adjusts LMs' architectures for numerical reasoning through numerical tokenization (Han et al., 2022) or additional, numerical embeddings (Jin et al., 2021).

**Systematic Probes for Tables.** Tables have been utilized previously used to create probes for table grounding (Gupta et al., 2022b) or recasting non-NLI datasets (e.g. question-answering) to NLI (Jena et al., 2022). Unlike unstructured text data, tables have a natural structure that allows creating controlled experiments more easily (Gupta et al., 2022a). We drew inspiration from prior tabular probing approaches and extended them for automating probing of numerical tabular data. Jena et al. (2022) introduce a generic approach to adjust

table QA datasets and generate NLI data. For data recasting, they follow a systemic approach similar to ours. However, the focus is on transforming QA datasets, emphasizing the end-result (i.e. the NLI data) through data augmentation. They do not evaluate the presence of numerical data in their tables or consider numerical reasoning in the model evaluation phrase.

**Comparison to Prior Work.** All the above mentioned prior works on numerical reasoning have provided motivation for our research. However, their evaluations have focused on a narrow range of reasoning types and models. Most study only concentrated on one specific model such as T5 (Pal and Baral, 2021), GPT3 (Muffo et al., 2022), or BERT (Park et al., 2022). In contrast, our framework provides a comprehensive evaluation of numerical reasoning skills. We cover a wide spectrum of complexity levels, ranging from representation to complex reasoning. Moreover, we assess a variety of models with diverse architectures, sizes, and training settings for numerical reasoning.

# 7 Conclusion

This paper presents a framework for probing language models' numerical reasoning skills. We organise skills in a taxonomy and generate large-scale sets of probes covering more than ten numerical reasoning types. Using table NLI as a case study, we evaluate the numerical reasoning abilities of ten models. These models belong to the categories numerical LMs, tabular LMs, and few-/zero-shot LLMs. We discuss reasoning types that prove challenging for the probed models and explore promising directions for future research.

**Future Directions.** For certain (numerical) tasks, tool-augmented LMs equipped with capabilities such as calculators or code execution have been proven valuable (Mialon et al., 2023). However, certain tasks require implicit numerical reasoning which might not necessarily involve direct calculations based on numbers. For instance, classifying sentences that incorporate numbers in varied settings, like time indications (Feng et al., 2023), currencies or conversations (Macina et al., 2023). Such tasks demand a numerical interpretation beyond mere arithmetic computations. Moreover, calling external tools using LMs requires basic numerical comprehension to invoke an external tool correctly (Chen et al., 2022).

## Limitations

This work proposes a taxonomy and framework to probe numerical reasoning skills in LMs. It involves the creation of large-scale probing sets using an automated approach. However, the evaluation of this approach is currently limited to the task of table NLI. For future research, it is interesting to extend this to include additional tasks and datasets. This extension serves two purposes: first, it allows evaluating a more diverse range of datasets. Second, it enables including challenges specific to other tasks.

In this paper, the evaluation of most reasoning types primarily involves structural changes at the hypotheses level. While we include counterfactual table probes, they are limited to one dataset and perturbations method only. Further research is needed to study models' performance on numerical data in the premise data. Therefore, we need table-based probes for all reasoning types of the proposed taxonomy.

## Ethics Statement

In this paper, we study the numerical reasoning skills of different LMs. However, to deploy these systems in real-world applications, further studies and evaluations specific to the intended use cases are required. In order to support future research, we plan to release all the scripts and resources used for probe creation and model evaluation. This will facilitate and encourage further research in this field.

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

## A  Insights

**Main Insights.** We investigated the language models and found that LLMs like FlanT5 and GPT3.5 perform better than other models on various numerical reasoning tasks. When the labels are switched around and when dealing with negative values, we found that both table-based and numerical models had difficulty comprehending the data. In contrast, DeBERTa performs relatively well compared to models like LUNA and PASTA, which are tuned for improved numerical reasoning skills.

## B  Probe Statistics

| Reasoning Type | Count |
|---|---|
| Word Problems | 238 |
| Sorting | 379 |
| Counterfactual | 1,000 |
| Currency | 1,014 |
| Negative | 3,316 |
| Range | 4,208 |
| Scientific notation | 6,274 |
| Arithmetic | 8,082 |
| Ordinal | 10,569 |
| Percentage | 16,851 |
| Date | 18,642 |
| Approximation | 20,440 |
| Comparison | 30,763 |
| Numeration | 166,319 |
| Total | 288,095 |
| Flipped probes | 77,687 |

Table 5: Breakdown of probes per reasoning type.

Table 2 gives an overview of probes per dataset. Most probes (i.e. $214,440$) are created from Tab-Fact hypotheses as this is also the biggest dataset available, followed by InfoTabs ($19,779$). Table 5 provides a breakdown of probes per reasoning type. In total, we have $286,857$ probes, of which $76,404$ are label-flipping probes.

In the ideal scenario with counterfactual tables, the models' performance should be similar to the performance on the original tables. However, we observed that TAPAS and DeBERTa's performance

improved significantly, which leads to the conclusion that models are biased toward one label.

Overall no language model excels in all the numerical reasoning tasks. Surprisingly, models perform relatively well in complex tasks like Numerical Word Problems but struggle at simple reasoning tasks like numeration and comparison.

| Model | Table Specific | | | Numerical Specific | | | | Large LMs | | | | | |
|---|---|---|---|---|---|---|---|---|---|---|---|---|---|
| | TAPAS | DeBERTa | TAPEX | NT5 | LUNA | PASTA | ReasTAP | FlanT5 | | GPT3.5 | | PaLM | |
| **Reasoning** | | | | | | | | few | zero | few | zero | few | zero |
| **Representation** | | | | | | | | | | | | | |
| Numeration | 19.02 | 61.69 | 74.83 | 64.19 | 63.57 | 85.43 | 75.35 | 64.59 | 67.58 | 77.6 | 61.83 | 76.38 | 70.76 |
| Heterogeneous | 79.25 | 56.61 | 81.17 | 66.85 | 70.71 | 89.11 | 76.49 | 70.17 | 71.04 | 78.82 | 61.26 | 76.75 | 69.85 |
| Negative | 59.38 | 69.81 | 97.01 | 51.77 | 24.66 | 70.44 | 81.04 | 70.8 | 20.37 | 85 | 87 | 92 | 88 |
| **Label Flipped** | | | | | | | | | | | | | |
| Numeration | 23 | 61.92 | 75.79 | 71.98 | 76.34 | 86.96 | 73.64 | 88.93 | 66.93 | 90.57 | 54.11 | 85.97 | 87.91 |
| Heterogeneous | 67.66 | 53.12 | 82.96 | 62.94 | 77.57 | 83.75 | 68.37 | 87.48 | 51.55 | 89.18 | 65.32 | 86.55 | 79.14 |
| Negative | 59.38 | 69.81 | 97.01 | 51.77 | 24.66 | 70.44 | 72.53 | 70.8 | 20.37 | 88 | 85 | 92 | 88 |
| **Number Sense** | | | | | | | | | | | | | |
| Scale | 74.11 | 67.29 | 78.32 | 63.05 | 72.24 | 77.82 | 31.1 | 61.25 | 69.59 | 79.17 | 69 | 78.04 | 71.15 |
| Comparison | 68.43 | 68.7 | 54.12 | 66.85 | 69.19 | 84.75 | 58.46 | 37.77 | 23.87 | 83.31 | 50.58 | 77.43 | 63.78 |
| Approximation | 75.49 | 64.65 | 71.64 | 63.3 | 67.09 | 86.08 | 58.66 | 62.28 | 78.09 | 75.13 | 59.62 | 76.88 | 75.35 |
| Range | 70.84 | 57.53 | 94.99 | 58.14 | 48.86 | 73.25 | 87.77 | 80.2 | 20.09 | 90 | 85.5 | 80 | 81.41 |
| **Label Flipped** | | | | | | | | | | | | | |
| Scale | 72.33 | 76.74 | 71.78 | 65.17 | 78.8 | 87 | 95.92 | 85.71 | 59.6 | 92.62 | 60.71 | 79.58 | 80.21 |
| Comparison | 53.55 | 70.75 | 38.29 | 68.64 | 69.76 | 85.74 | 42.69 | 86.12 | 49.66 | 96.05 | 37.34 | 82 | 54.86 |
| **Manipulation** | | | | | | | | | | | | | |
| Sorting | 68.71 | 49.98 | 94.65 | 69.83 | 51.11 | 69.87 | 83.33 | 75.44 | 61.4 | 91.44 | 77 | 68.94 | 70.5 |
| Arithmetic | 71.15 | 58.01 | 95.92 | 58.83 | 73.6 | 73.6 | 88.97 | 78.92 | 22.42 | 89.5 | 85.5 | 83.5 | 81.5 |
| **Complex Reasoning** | | | | | | | | | | | | | |
| Complex | 54.6 | 52.65 | 82.69 | 52.94 | 39.22 | 81.93 | 95.92 | 91.69 | 69.03 | 93 | 56.1 | 94 | 91.3 |
| Counterfactual | 59.71 | 43.23 | 88.7 | 80.98 | 81.3 | 83.52 | 86.9 | 35.46 | 62.5 | 58.18 | 69.09 | 55.38 | 53.85 |

Table 6: Results on original sets (average accuracy).

| Model | Table Specific | | | Numerical Specific | | | | Large LMs | | | | | |
|---|---|---|---|---|---|---|---|---|---|---|---|---|---|
| | TAPAS | DeBERTa | TAPEX | NT5 | LUNA | PASTA | ReasTAP | FlanT5 | | GPT3.5 | | PaLM | |
| **Reasoning** | | | | | | | | few | zero | few | zero | few | zero |
| **Representation** | | | | | | | | | | | | | |
| Numeration | 18.4 | 59.24 | 68.34 | 60.5 | 59.44 | 79.64 | 69.36 | 64.5 | 63.01 | 77.19 | 64.3 | 75.38 | 74 |
| Heterogeneous | 72.76 | 54.52 | 75.75 | 62.19 | 63.73 | 80.59 | 74.32 | 68.92 | 67.52 | 77.36 | 65.78 | 75.5 | 70.22 |
| Negative | 32 | 60.2 | 97.55 | 2.86 | 6.03 | 62.92 | 76.19 | 84.4 | 28.98 | 92 | 89 | 90 | 89 |
| **Label Flipped** | | | | | | | | | | | | | |
| Numeration | 10.13 | 43.68 | 39.41 | 33.89 | 29.82 | 53.3 | 41.05 | 17.34 | 70.04 | 56.83 | 73.37 | 52.54 | 45.91 |
| Heterogeneous | 63.1 | 42.45 | 49.44 | 57.08 | 47.97 | 68.75 | 56.58 | 40.57 | 80.34 | 62.16 | 78.83 | 70.12 | 62.18 |
| Negative | 20.9 | 40.8 | 2.9 | 42.52 | 43.61 | 20.72 | 41.28 | 11.38 | 55.63 | 32 | 86.87 | 18 | 19 |
| **Number Sense** | | | | | | | | | | | | | |
| Scale | 71.68 | 49.43 | 68.38 | 60.39 | 62.21 | 73.63 | 67.23 | 53.12 | 62 | 80.46 | 66 | 75.15 | 59.39 |
| Comparison | 42.78 | 54.91 | 45.32 | 45.94 | 54.79 | 60.04 | 40.73 | 48.84 | 55.33 | 73.74 | 58.69 | 64.97 | 51.71 |
| Approximation | 62.94 | 58.65 | 56.01 | 56.75 | 61.56 | 61.55 | 51.26 | 56.23 | 67.65 | 74.96 | 66.1 | 69.25 | 66.4 |
| Range | 57.73 | 38.09 | 93.13 | 46.28 | 6.46 | 11.24 | 92.12 | 98.2 | 55.87 | 90.47 | 84.5 | 91.5 | 85 |
| **Label Flipped** | | | | | | | | | | | | | |
| Scale | 39.26 | 44.24 | 38.91 | 44.55 | 28.63 | 43.12 | 98.46 | 23.69 | 77.67 | 56.31 | 66.94 | 60.93 | 52.57 |
| Comparison | 62.59 | 57.28 | 37.15 | 46.9 | 66.75 | 69.72 | 34.74 | 66.43 | 50.29 | 75.22 | 54.6 | 69.43 | 44.98 |
| **Manipulation** | | | | | | | | | | | | | |
| Sorting | 44.8 | 63.75 | 8.47 | 53.89 | 77.79 | 64.54 | 13.29 | 10.03 | 76.75 | 38.75 | 72.81 | 46.38 | 42.31 |
| Arithmetic | 71.15 | 43.52 | 4.28 | 42.89 | 37.5 | 37.5 | 9.88 | 22.45 | 81.08 | 35.5 | 89.5 | 27 | 28.5 |
| **Complex Reasoning** | | | | | | | | | | | | | |
| Complex | 89.22 | 56.3 | 16.39 | 51.24 | 55.46 | 40.28 | 3.4 | 9.38 | 41.42 | 37 | 72.78 | 25 | 22 |
| Counterfactual | 86.21 | 67.24 | 77.8 | 31.23 | 82.1 | 78.43 | 78.1 | 57.27 | 70.14 | 81.82 | 72.73 | 72.31 | 80 |

Table 7: Results on probed sets (average accuracy).