# OpenReview forum: "Exploring the Numerical Reasoning Capabilities of Language Models: A Comprehensive Analysis on Tabular Data"
_EMNLP/2023/Conference — EMNLP 2023 Findings_

### Official Review · Reviewer_CcwK · 2023-07-21

**Soundness:** 3

**Excitement:**

2: Mediocre: This paper makes marginal contributions (vs non-contemporaneous work), so I would rather not see it in the conference.

**Paper Topic And Main Contributions:**

This paper presents a new taxonomy for numerical reasoning skills at four levels, namely, representation, number sense, manipulation, and complex reasoning encompassing over 10 reasoning types. Then, they propose an approach to create a probing dataset for each reasoning type using Tabular Natural Language Inference datasets and analyze performance of several models such as FlanT5, GPT 3.5, NT5, and DeBERTa.

**Reasons To Accept:**

1. This paper investigates numerical reasoning which has been and still is one of the toughest skills for language models as they often struggle to reason correctly over the numerical information in text.

2. This paper presents an approach to create probing data for different numerical reasoning skills which may help in doing fine-grained investigations of LLM’s basic understanding of numbers.

**Reasons To Reject:**

1. Firstly, the work lacks sufficient contributions for it to be a long paper. Since, most of the strategies to create probing data in section 4 are about replacing the numbers, dates, etc. I think it can be presented in a much more concise manner. Also, previous works such as Thawani et al., 2021b; Xu et al., 2022, DROP dataset have proposed taxonomies and datasets that cover majority of the skills presented in this work such as Numeration, Arithmetic, Comparison, Word Problems, Measurement Estimation, DROP includes Comparison, Subtraction, Selection, Addition, Counting, Sorting, Coreference resolution, Word Problems, etc. Thus, section 2 can also be concisely presented.

2. The need for a new taxonomy is not well motivated in the paper as previous works such as Thawani et al., 2021b; Xu et al., 2022 already provide a detailed taxonomy covering numerical reasoning skills.

3. In the majority of Section 5.3, just the numbers from Table 4 have been mentioned in words. It lacks conclusive analysis. Also, this work doesn’t lead to new results or findings as such.

Furthermore, the paper “Leveraging data recasting to enhance tabular reasoning” uses more techniques to create Tabular NLI instances. It uses augmented instances to improve reasoning; in this work, a simpler augmentation technique is used for creating probing data. This paper and how it is different from the current work needs to be discussed.

**Reproducibility:**

3: Could reproduce the results with some difficulty. The settings of parameters are underspecified or subjectively determined; the training/evaluation data are not widely available.

**Reviewer Confidence:**

3: Pretty sure, but there's a chance I missed something. Although I have a good feel for this area in general, I did not carefully check the paper's details, e.g., the math, experimental design, or novelty.

**Typos Grammar Style And Presentation Improvements:**

L067 -  four level:  →  four levels:

L260 - Overall, we our framework  → Overall, our framework

L291 - This section discussed probes → This section discusses probes

L296 - Onw converting  → One converting

L427 → which are divers in terms  → which are diverse in terms

...

---

> ### Author Rebuttal · Authors · 2023-08-29
>
> We appreciate the time and effort the review dedicated to reviewing our paper and thank you for the questions.
>
> **Regarding comparison to previous works: taxonomy and insights gained**
>
> While the mentioned works indeed provide important insights on numerical reasoning, our work fills a gap by introducing and examining a taxonomy that captures reasoning types across four different levels including eleven different reasoning types (Figure 1). Moreover, our work allows automated creation of probes using tabular data, a data source which has a high prevalence of numerical data.
>
> We thank the author for highlighting the work by Thawani et al. (2021) [1], Xu et al. (2022) [2] and the DROP paper, while we include the references in our paper, we will extend this with a more detailed comparison in the related work section as the reviewer suggests.
>
> While **Thawani et al.** [1] address the broader issues of number representation in NLP systems, our work dives into more specific aspects of numerical reasoning. Hereby, we use tabular data to create probes capturing these desired reasoning types. Tables inherently have a structured form of representing numerical data, where numbers don't stand alone but relate to other numbers and categories in a distinct manner.
> Moreover, Thawani et al. [1] offer a taxonomy based primarily on granularity (exact vs. approximate) and units (abstract vs. grounded) of number representation. While we acknowledge the significance of their taxonomy, our framework offers an enhanced taxonomy that captures representation of numbers, number sense (i.e. the meaning of numbers in the given context), manipulation of numbers, and complex reasoning with numerical data. The taxonomies, while having some overlaps, serve different purposes.
> Finally, while Thawani et al. [1] touch upon the concept of probing, our probing techniques are tailored to evaluate the model's reasoning capabilities over numerical data within tables. Their work makes an important contribution on surveying and categorizing previous work on number representation. Our probing evaluates models on their ability to understand numbers and reason with them in the context of structured tabular data.
>
> **Xu et al. [2]** is also an important work on numerical reasoning with LMs. Hence, we will include the paper in our related work and compare it in more detail to our work (e.g. on the taxonomy and findings).
> Next, we provide some details on this comparison. This and additional points will be added to the paper.
> *Robustness vs. reasoning*: Xu et al. [2] focus on the robustness of NLP models in handling numerical data which is a very important constitution to the NLP and numeracy. Our probing study on the other side pivots towards the reasoning capabilities of models when dealing with numerical and tabular data. This means we delve deeper into understanding how well models can make sense of numerical relations in a structured format rather than just testing their resilience.
> *Comparing reasoning types*: [2] gives interesting insights on models reasoning on two levels: a) number detection and extraction; b) semantic parsing of numbers
> a) is comparable to our reasoning types "numeration". Similar to us, they evaluate number mapping between numerals and number tokens. They also include float numbers while studying number detection. We take a similar approach for reasoning type "heterogeneous numbers" but additionally include date, fraction, percentage, scientific notation as heterogeneous number types in addition to floats. These number representations are commonly found in various real-world documents.
> b) is similar to our reasoning type "arithmetic". However while this work gives interesting insights on number parsing and using operands for calculations based on parsed numbers, our probes give a broader number of reasoning types and related challenges. For example, understanding the value of numbers by means of comparison between them, sorting numbers to understand their relation to each other, and so on.
> Additionally, our work also includes further reasoning types, which we motivate and discuss in detail in our paper.
>
> Comparing out work to the **DROP** benchmark [4], would like to highlight the following aspects that we will include in the revised paper as well:
> The DROP paper [4] focuses on a diversified set of reasoning types that revolve largely around discrete reasoning over paragraphs. These reasoning types require a combination of natural language understanding and simple arithmetic.
> On the other hand, our work categorizes numerical reasoning more granularly, specifically emphasizing language models' capability to reason with numbers in various contexts. The taxonomy in our paper spans over ten reasoning types, some of which encompass the reasoning types in DROP but go deeper in terms of complexity and specificity. For instance, while DROP emphasizes basic arithmetic like addition and subtraction, probes more intricate numerical reasoning types that might not be solely arithmetic-based but hinge on understanding numbers themselves (representation and number sense) and in different numerical contexts.
> Moreover, a distinguishing feature of our work is the usage of table-based Natural Language Inference (NLI) to understand and evaluate the numerical reasoning of models. This is crucial as tables represent structured data are prevalent in the real-world and using our probing approach allow creating probes on a large scale.
> While both works share a thematic focus on numerical reasoning, the methodology employed in our paper offers a distinct exploration of models' numerical capabilities, especially in the context of structured data like tables.
>
> **Regarding comparison to Jena et al. (2022) [3]**
>
> Thank you for highlighting this work by Jena et al. (2022) [3] While we refer to this work for probe creation and in the related work section of the paper, we acknowledge the feedback and will clarify with a more detailed comparison in the related work section.
> Our research focuses on probing the numerical reasoning capacities of language models, introducing a taxonomy to classify various reasoning types and employing table-based Natural Language Inference (NLI) to evaluate different model categories. In contrast, Jena et al. (2022) [3] concentrate on data transformation, applying data recasting to generate NLI datasets from tabulated structured data for QA. The aim of their work is to enrich the NLI dataset landscape.
> Moreover, while we use tables for numerical reasoning, they use tables to transform questions and answers into NLI statements, emphasizing the process and end-result (i.e. the NLI data), rather than the reasoning behind the answers.
> Our evaluation assesses diverse models on numerical reasoning challenges included in our taxonomy. Jena et al., (2022) [3] however, assess the worth of their recast data based on its efficacy in both augmenting and evaluating tabular inference tasks, showing the utility of their generated datasets in real-world applications.
>
> **Regarding insights of our work**
>
> Below we highlight the most important insights gained through our probing study (as described in Section 5 of the paper). We appreciate the reviewers feedback and highlight the main findings more clearly in the paper.
>
> *Model Performance Across Numerical Reasoning Types:*
> No singular model emerged as dominant across all types of numerical reasoning. However, the models FlanT5 and GPT3.5 showcased commendable numerical reasoning skills. GPT3.5, in particular, exhibited significant resilience, except when faced with complex reasoning probes. This might stem from its vast training on the extensive 300 TB Common Crawl data, allowing it a broader numerical data exposure than others. Notably, FlanT5's fine-tuning, emphasizing instructions and chain-of-thought reasoning, enabled it to outshine NT5, even though both have the T5 model as their foundation.
>
> *Representation and Number Sense:*
> When it came to representation, models TAPAS and FlanT5 exhibited superior performance, particularly in non-flipping numeration probes. However, a performance dip was observed when models encountered negative number probes, suggesting potential challenges in recognizing the "-" sign representation of negative numbers. Yet, few-shot models like FlanT5 and GPT3.5 seemed more adept at comprehending the linguistic expression of negatives. Regarding number sense, while fine-tuned models found comparison probes challenging, FlanT5 and GPT3.5 managed to keep their performance, sometimes even outperforming their base hypotheses.
>
> *Manipulation, Complex Reasoning, and Label Flipping:*
> In the categories of manipulation and complex reasoning, fine-tuned models generally saw a 23.5% average decline in accuracy, with LUNA being an exception. Surprisingly, few-/zero-shot models improved their performance by 9.5%. Yet, when confronted with complex reasoning challenges, fine-tuned models seem to perform better in general. Label flipping, however, posed a significant challenge. A steeper performance drop in flipping probes as compared to their non-flipping counterparts highlighted models' potential over-reliance on spurious patterns, rather than genuine comprehension.
>
> *Numerical vs. Table-Specific Models:*
> LUNA, which utilized a unique tokenization approach for numbers, showed similar performance to other models across various reasoning types, except for a slight edge in comparison flipping probes. PASTA, despite its numeric-focused design and objectives, only outperformed DeBERTa in handling negative number and scale probes. TAPAS, a table-based model, performed well non-flipping scenarios but showed declines when labels were flipped, emphasizing its challenges with spurious correlations.
>
> In general, our probing evaluation shows that performance variances across different models for numerical reasoning, underscoring the need for detailed evaluation of models with fine-grained categories of numerical reasoning to understand their strengths and weaknesses in detail.
>
> For the final paper we further plan to extend our three model categories with one additional model per category. Concretely, TAPEX and ReasTAP [5] which we have started probing over the last days.
>
> **References**
>
> [1] Thawani, Avijit, et al. "Representing Numbers in NLP: a Survey and a Vision." Proceedings of the 2021 Conference of the North American Chapter of the Association for Computational Linguistics: Human Language Technologies. 2021.
>
> [2] Jialiang Xu, Mengyu Zhou, Xinyi He, Shi Han, and Dongmei Zhang. 2022. Towards robust numerical question answering: Diagnosing numerical capabilities of NLP systems. EMNLP 2022.
>
> [3]  Jena, Aashna, et al. "Leveraging Data Recasting to Enhance Tabular Reasoning." Findings of the Association for Computational Linguistics: EMNLP 2022.
>
> [4] Dua, Dheeru, et al. "DROP: A Reading Comprehension Benchmark Requiring Discrete Reasoning Over Paragraphs." Proceedings of the 2019 Conference of the North American Chapter of the Association for Computational Linguistics: Human Language Technologies, Volume 1 (Long and Short Papers). 2019.
>
> [5] Zhao, Yilun, et al. "ReasTAP: Injecting Table Reasoning Skills During Pre-training via Synthetic Reasoning Examples." Proceedings of the 2022 Conference on Empirical Methods in Natural Language Processing. 2022.

---

### Official Review · Reviewer_FvRi · 2023-08-04

**Soundness:** 4

**Excitement:**

4: Strong: This paper deepens the understanding of some phenomenon or lowers the barriers to an existing research direction.

**Missing References:**

no

**Paper Topic And Main Contributions:**

This paper assesses models on numerical reasoning abilities in a systematic way. It proposes a hierarchical taxonomy for numerical reasoning skills. And it probes existing models systematically. It is inspiring for future works on this topic.

**Questions For The Authors:**

1. what about the statistics aspect of the number beyond range, e.g., mean, sum, correlation etc.


**Reasons To Accept:**

1. The topic is important. I have not seen similar empirical studies on this topic before. It's promising to have such fine-grained study.
2. The paper conducts a thorough evaluation of state-of-the-art models on all reasoning types, using a diverse and extensive set of numerical probes and measuring performance shifts.
3. The experiment results and analyses are inspiring for future works. According to the study, there is not one best-performing model across all reasoning types, so it will motivate future work on unifying the advantages of different methods.

**Reasons To Reject:**

1. Benchmarks of numerical reasoning datasets can be extended, e.g., GSM8K, FinQA, HiTab, DROP, etc.
2. More table-specific models can be explored like TaBERT, TURL, TUTA, etc

**Reproducibility:**

4: Could mostly reproduce the results, but there may be some variation because of sample variance or minor variations in their interpretation of the protocol or method.

**Reviewer Confidence:**

4: Quite sure. I tried to check the important points carefully. It's unlikely, though conceivable, that I missed something that should affect my ratings.

**Typos Grammar Style And Presentation Improvements:**

-

---

> ### Author Rebuttal · Authors · 2023-08-29
>
> We thank the reviewers for their input and are excited to read they found our work to make a valuable contribution to an important topic.
>
> **Regarding additional benchmarks on numerical reasoning**
>
> We thank the reviewer for highlighting additional numerical reasoning datasets to extend our work. Following previous work on table-based probing/recasting frameworks (Gupta et al., 2022; Jena et al., 2022) [1,2], we concentrated on five datasets (see Table 2) that capture various types of numerical reasoning skills. In the future, we plan to extend this work with additional datasets, models, reasoning types, and probing strategies (e.g., table-based probes). We will include some of this in the revised final paper version (see our replies to other reviewers) and others in future work.
>
> **Regarding probing additional models**
>
> Our goal on the model level was to cover a broad range of models. Hence, we included at least two models from three different categories (see Sec 5.1 for more details):
>
> 1. Numerical LMs: LUNA and PASTA (both are tabular reasoning models focusing on numerical data), NT5
> 2. LMs for tabular reasoning: TAPAS and DeBERTa with a table linearization approach
> 3. LLMs: FlanT5, GPT3.5 (to include both an open source model, i.e., FlanT5, and a closed model with high performance on various tasks, i.e., GPT3.5)
>
> However, we value that the author mentioned including other Tabular models. For the final paper, we plan to extend our three model categories with one additional model per category. Concretely, TAPEX and ReasTAP [4], which we have started probing over the last few days.
>
> **References**
>
> [1] Gupta, Vivek, et al. "Is my model using the right evidence? systematic probes for examining evidence-based tabular reasoning." Transactions of the Association for Computational Linguistics 10 (2022): 659-679.
>
> [2] Jena, Aashna, et al. "Leveraging Data Recasting to Enhance Tabular Reasoning." Findings of the Association for Computational Linguistics: EMNLP 2022.
>
> [3] Eisenschlos, Julian, Syrine Krichene, and Thomas Mueller. "Understanding tables with intermediate pre-training." Findings of the Association for Computational Linguistics: EMNLP 2020. 2020.
>
> [4] Zhao, Yilun, et al. "ReasTAP: Injecting Table Reasoning Skills During Pre-training via Synthetic Reasoning Examples." Proceedings of the 2022 Conference on Empirical Methods in Natural Language Processing. 2022.

---

### Official Review · Reviewer_pyHU · 2023-08-06

**Typos Grammar Style And Presentation Improvements:** Can the full version of Table 4 be pr…
**Soundness:** 3

**Excitement:**

3: Ambivalent: It has merits (e.g., it reports state-of-the-art results, the idea is nice), but there are key weaknesses (e.g., it describes incremental work), and it can significantly benefit from another round of revision. However, I won't object to accepting it if my co-reviewers champion it.

**Paper Topic And Main Contributions:**

This paper presents a method for analyzing the performance of language models in terms of numerical reasoning. The authors study a wide variety of models such as DeBERTa, NT5, FlanT5 / GPT3.5. The authors define a taxonomy of reasoning types with corresponding datasets build upon previously existing ones.

Update after response: Soundness increased to 3.

**Questions For The Authors:**

* How do you view tool-use based methods in the context of your proposed evaluation framework?

**Reasons To Accept:**

The paper presents an interesting perspective on how language models represent numbers and how well they are able to reason about them.
* The paper presents results with a variety of models, this provides an interesting angle on how these models perform.
* The authors motivate the task and describe their proposed datasets / evaluation methods thoroughly.
* Benchmarks and standards for evaluating future models on this task.

**Reasons To Reject:**

* I find the table, Table 4, is a bit hard to read. Why not present the actual accuracies? I also think that it's a bit hard for the reader to understand some of the large performance swings in the models
* The answer to Q1 seems to be in the negative. Yet, I feel that the hypotheses explored could be expanded to include a wider variety of analysis. I think devoting more of the body of the paper to analysis of Q1-Q5 would benefit the paper, perhaps removing content from earlier sections.
* I think that it would be nice to consider the role of facts stored in the parameters of LLMs a bit more. E.g., for some of the tables, I think that the unaltered tables could contain things that the model has memorized. I think that understanding how this changes model behavior could be interesting.

**Reproducibility:**

2: Would be hard pressed to reproduce the results. The contribution depends on data that are simply not available outside the author's institution or consortium; not enough details are provided.

**Reviewer Confidence:**

3: Pretty sure, but there's a chance I missed something. Although I have a good feel for this area in general, I did not carefully check the paper's details, e.g., the math, experimental design, or novelty.

---

> ### Author Rebuttal · Authors · 2023-08-29
>
> We thank the reviewers and are encouraged they found our work to be an interesting perspective on how LMs represent numbers and reason about them.
>
> **Regarding accuracy tables.:**
> We thank you for raising this point. While we initially included accuracy, we swapped it by percentage to allow better comparison of performance changes across reasoning types and models. We do understand and value the point raised by the review. Therefore, we will include accuracy tables in the final version of the paper. If space allows, we will include tables in the main paper. Otherwise, we will split the tables between the main and appendix.
>
> Below, please find the full table for *numeration (category representation)* as an example, and similar tables will be added for other reasoning types as well.
>
> | dataset  | split       | type    | label flip | model         | original accuracy | probe accuracy | delta          |
> | -------- | ----------- | ------- | ---------- | ------------- | ----------------- | -------------- | -------------- |
> | InfoTabs | alpha 1     | int2str | no         | FlanT5        | 61.95             | 61.64          | \-0.5004035513 |
> | InfoTabs | alpha 1     | int2str | yes        | FlanT5        | 98.13             | 3.75           | \-96.17853867  |
> | InfoTabs | alpha 2     | int2str | no         | FlanT5        | 62.1              | 59.24          | \-4.60547504   |
> | InfoTabs | alpha 2     | int2str | yes        | FlanT5        | 95.45             | 5.84           | \-93.88161341  |
> | InfoTabs | alpha 3     | int2str | no         | FlanT5        | 70.07             | 66.67          | \-4.852290567  |
> | InfoTabs | alpha 3     | int2str | yes        | FlanT5        | 96.51             | 8.85           | \-90.82996581  |
> | infotabs | test_alpha2 | int2str | no         | TAPAS         | 48.73             | 47.77          | \-1.97003899   |
> | infotabs | test_alpha3 | int2str | no         | TAPAS         | 68.27             | 64.93          | \-4.892339241  |
> | infotabs | test_alpha1 | int2str | yes        | TAPAS         | 71.59             | 20.45          | \-71.4345579   |
> | infotabs | test_alpha3 | int2str | yes        | TAPAS         | 62.79             | 38.76          | \-38.27042523  |
> | infotabs | test_alpha1 | int2str | no         | TAPAS         | 68.34             | 66.46          | \-2.750951127  |
> | infotabs | test_alpha2 | int2str | yes        | TAPAS         | 51.14             | 37.5           | \-26.67188111  |
> | infotabs | dev         | int2str | yes        | TAPAS         | 84.21             | 21.93          | \-73.95796224  |
> | infotabs | test_alpha1 | int2str | no         | deberta_large | 78.93             | 74.62          | \-5.460534651  |
> | infotabs | dev         | int2str | yes        | deberta_large | 85.96             | 36.84          | \-57.14285714  |
> | infotabs | test_alpha3 | int2str | no         | deberta_large | 69.28             | 67.62          | \-2.396073903  |
> | infotabs | test_alpha3 | int2str | yes        | deberta_large | 59.69             | 46.51          | \-22.08075054  |
> | infotabs | test_alpha1 | int2str | no         | GPT3.5        | 0.75              | 0.75           | 0              |
> | infotabs | test_alpha3 | int2str | no         | GPT3.5        | 0.76              | 0.75           | \-1.315789474  |
> | infotabs | test_alpha2 | int2str | no         | GPT3.5        | 0.66              | 0.6767676768   | 2.540557086    |
> | infotabs | test_alpha2 | int2str | yes        | LUNA_TAPAS    | 67.05             | 26.14          | \-61.01416853  |
> | infotabs | dev         | int2str | yes        | LUNA_TAPAS    | 91.23             | 11.4           | \-87.50411049  |
> | infotabs | test_alpha3 | int2str | no         | LUNA_TAPAS    | 71.61             | 64.93          | \-9.328306102  |
> | infotabs | test_alpha3 | int2str | yes        | LUNA_TAPAS    | 77.52             | 24.81          | \-67.99535604  |
> | infotabs | test_alpha2 | int2str | no         | LUNA_TAPAS    | 55.41             | 48.41          | \-12.63309872  |
> | infotabs | test_alpha1 | int2str | yes        | LUNA_TAPAS    | 85.23             | 6.82           | \-91.99812273  |
> | infotabs | test_alpha1 | int2str | no         | LUNA_TAPAS    | 72.41             | 66.77          | \-7.788979423  |
> | infotabs | test_alpha1 | int2str | yes        | nt5           | 76.14             | 27.27          | \-64.18439716  |
> | infotabs | test_alpha2 | int2str | yes        | nt5           | 53.41             | 53.41          | 0              |
> | infotabs | test_alpha2 | int2str | no         | nt5           | 50                | 49.72          | \-0.56         |
> | infotabs | test_alpha1 | int2str | no         | nt5           | 62.94             | 57.61          | \-8.468382587  |
> | infotabs | dev         | int2str | yes        | nt5           | 73.68             | 34.21          | \-53.56948969  |
> | infotabs | test_alpha3 | int2str | no         | nt5           | 51.81             | 49.4           | \-4.651611658  |
> | infotabs | test_alpha3 | int2str | yes        | nt5           | 48.84             | 41.86          | \-14.29156429  |
> | infotabs | test_alpha1 | int2str | yes        | PASTA_DeBERTa | 92.05             | 44.32          | \-51.85225421  |
> | infotabs | test_alpha2 | int2str | yes        | PASTA_DeBERTa | 77.27             | 56.82          | \-26.46563996  |
> | infotabs | test_alpha2 | int2str | no         | PASTA_DeBERTa | 76.43             | 73.57          | \-3.741986131  |
> | infotabs | test_alpha1 | int2str | no         | PASTA_DeBERTa | 85.58             | 82.45          | \-3.657396588  |
> | infotabs | dev         | int2str | yes        | PASTA_DeBERTa | 88.6              | 49.12          | \-44.55981941  |
> | infotabs | test_alpha3 | int2str | no         | PASTA_DeBERTa | 83.09             | 81             | \-2.515344807  |
> | infotabs | test_alpha3 | int2str | yes        | PASTA_DeBERTa | 75.97             | 48.84          | \-35.71146505  |
>
> **Regarding models’ performance swing**
> The large performance swings mainly occur due to inconsistent numerical reasoning of models across tasks. For example, we observe that some models struggle with more basic reasoning (e.g., FlanT5 zero on numeration) while performing very well on more complex types.
>
> This behavior might have different reasons, which we will include and discuss in the final paper version to provide more insights (beyond what is addressed with Q1-Q4 currently). One potential reason is memorization. Previous work [1,2,3] on LMs and memorization shows that models tend to store knowledge in their parameters, which are retrieved during a task instead of reasoning with the provided input. Hence, models can memorize common arithmetic operations during their pretraining phase and, therefore, perform very well on certain tasks. For example, flipping numbers in words (“two”) to numerics (“2”) might allow models to retrieve knowledge which they didn't consider for the initial hypothesis. Therefore, we tried to design our probes as diverse as possible (e.g., “2” to “two” flipping and vice versa).
>
> A reason for very high-performance drops can be the hallucination of models, as discussed by Gupta et al. (2022) [4]. While models might have initially performed very well on the hypothesis, adjusting the numbers can hinder models from relying on spurious patterns. We will include this and further discussion on model performance swings in the final paper version in section 5 as a newly introduced question (i.e., Q5).
>
> **Regarding Q1 being negative and expanding the hypotheses studied in Section 5**
>
> This is a correct observation. As discussed earlier, the results show that the studied models show inconsistent performance on numerical reasoning skills. For different studied types, different models perform well.
>
> Thanks for the suggestion to expand the discussion. We will expand the discussion to additional insights as we will have an additional page in the final paper version. Among others, we plan to include the following points (which we also highlighted to another reviewer) for the different hypotheses (Q1-Q4) already included in the paper.
>
> Q1: The question of whether there is one model excelling in all reasoning types will be extended with further discussion on the strengths and weaknesses of each model and how this might be related to the model’s architecture, training procedure, and evaluation setting (for example, the differences we see between finetuned, few-, and zero-shot models)
>
> Q2: This question discusses results per reasoning type. While we already split the discussion into the different reasoning types (representation, number sense, etc.), additional space will allow us not only to discuss the categories of reasoning types but each reasoning type among them individually. For example, for the category “representation,” we find differences between numeration/heterogeneous and the other reasoning types in this category. This can be related to the point that numeration/heterogeneous probes mostly don't have changes in meaning. Whereas changing a number to its negative or flipping shows strong performance shifts in very high and very low accuracies compared to the hypothesis accuracy values.
>
> Q4: While we compare tabular and numerical models and discuss them, we will further include insights for the LLMs which were probed in few/zero-shot settings and more intra (not only inter) group differences and what the reasoning for the differences are.
>
> **Regarding knowledge stored in models’ parameters (i.e., memorization)**
>
> We thank the reviewer for raising this point! There is an interesting line of work looking into some of these aspects Gupta et al. (2022) [4]. This paper studies the phenomenon of memorization using tabular data. The paper alters the evidence tables for NLI datasets using different operations (e.g., row flipping, perturbation) to understand if (1.) models' results correlate to spurious patterns in the tables and (2.) they change their predictions if information in tables is altered.
>
> However, this work doesn't study this phenomenon specifically for numerical tasks.  In L54, we mention that our work focuses on studying numerical reasoning on the hypotheses level and that we plan to extend this in future work with additional evidence-level probing techniques to study phenomenon we observe potential table-related reasons for models’ high-performance changes (such as memorization, hallucination) in more detail.
>
> **Regarding tool-based augmented models in the context of our work**
>
> For certain tasks, Augmented Language Models (ALMs) equipped with capabilities such as calculators or code generation can be immensely valuable.
>
> However, not every task is best suited for such tools. For instance, there are tasks that require implicit numerical reasoning, which might not necessarily involve direct calculations based on numbers. A prime example of this is classifying sentences that incorporate numbers in varied settings, like time indications, dates, or currencies. Such tasks demand a nuanced understanding beyond mere numerical computation.
>
> Further, using LMs with equipped tools also requires a basic numerical comprehension on the part of the model. Without this understanding, invoking a tool correctly can become a challenge.
>
> We thank the reviewer for pointing this out to us. We will discuss ALMs in the context of our work (together with a few other points) in a separate section we plan to include before the “conclusion” named “future work, open challenges."
>
> **References**
>
> [1] Petroni, Fabio, et al. "Language Models as Knowledge Bases?." Proceedings of the 2019 Conference on Empirical Methods in Natural Language Processing and the 9th International Joint Conference on Natural Language Processing (EMNLP-IJCNLP). 2019.
>
> [2] Carlini, Nicholas, et al. "Extracting training data from large language models." 30th USENIX Security Symposium (USENIX Security 21). 2021.
>
> [3] Ishihara, Shotaro. "Training Data Extraction From Pre-trained Language Models: A Survey." arXiv preprint arXiv:2305.16157 (2023).
>
> [4] Gupta, Vivek, et al. "Is my model using the right evidence? systematic probes for examining evidence-based tabular reasoning." Transactions of the Association for Computational Linguistics 10 (2022): 659-679.

---

### Official Review · Reviewer_9VRp · 2023-08-11

**Soundness:** 4

**Excitement:**

3: Ambivalent: It has merits (e.g., it reports state-of-the-art results, the idea is nice), but there are key weaknesses (e.g., it describes incremental work), and it can significantly benefit from another round of revision. However, I won't object to accepting it if my co-reviewers champion it.

**Paper Topic And Main Contributions:**

With the help of a specially curated probe dataset, this paper evaluates the numerical reasoning capabilities of language models (LMs) in the tabular Natural Language Inference (TNLI) task. Specifically, samples from five tabular datasets are converted to the <tabular premise,hypothesis> format and LMs are required to classify if the hypothesis entails or contradicts the table. Moreover, three perturbations are introduced to create adversarial examples or probe examples, such as altering the hypothesis while maintaining the label unchanged. Finally, the authors evaluate three types of LMs with the resulted probe dataset and give detailed experimental analyses.

The contributions are two-folded. First, the authors introduce a hierarchical taxonomy for numerical reasoning skills and construct a probe dataset based on the proposed taxonomy. Secondly, the authors conduct extensive experiments to evaluate several LMs including SOTA LLMs such as FlanT5 and GPT-3.5.

**Reasons To Accept:**

(1) This paper focuses on the important numerical reasoning skills of LMs especially LLMs, which is a key necessity in many application scenarios.
(2) The authors introduces a new taxonomy for numerical reasoning skills together with a probe dataset for TNLI task.
(3) Comprehensive experiments are conducted across various LMs and several insightful findings are given.

**Reasons To Reject:**

(1) In consideration of previous work [1], the contribution of this paper seems to be incremental. [1] proposed a taxonomy for numerical reasoning skills and it also introduced pertubations to construct adversarial (probe) examples, which is very similar with this paper. For instance, changing "192" to "one hundred and ninety-two", which is the same as the "Numeration" probes. As a result, this paper looks like an imitation of [1] with incremental adaptation.

(2) Some recent sota LMs are not included in the experiments. For example, the authors choose TAPAS and DeBERTa as representatives of "LMs for tabular reasoning". However, new Table pre-training models such as TAPEX[2] were proposed and can not be treated as contemporary works.

[1] Jialiang Xu, Mengyu Zhou, Xinyi He, Shi Han, and Dongmei Zhang. 2022. Towards robust numerical question answering: Diagnosing numerical capabilities of NLP systems. EMNLP 2022.
[2] Liu Q, Chen B, Guo J, et al. TAPEX: Table pre-training via learning a neural SQL executor.

**Reproducibility:**

4: Could mostly reproduce the results, but there may be some variation because of sample variance or minor variations in their interpretation of the protocol or method.

**Reviewer Confidence:**

4: Quite sure. I tried to check the important points carefully. It's unlikely, though conceivable, that I missed something that should affect my ratings.

**Typos Grammar Style And Presentation Improvements:**

(1) In the caption of Table 1, "apprimation" --> " approximation". The label of "Date Flip" probe example of H1 should be "C".
(2) In section 2, "Figure 1" should be "Table 1" or "Table 3"?

---

> ### Author Rebuttal · Authors · 2023-08-29
>
> We thank the reviewers for their detailed review, feedback, and questions, which we address below in more detail. Moreover, we are delighted they find our work focusing on important numerical reasoning skills and providing comprehensive experiments and insightful findings.
>
> **Comparison to previous work, specifically [1]**
>
> The paper [1] is an important work on numerical reasoning with LMs. Hence, we will include the paper in our related work. We also plan to include a detailed comparison between their findings and ours.
>
> Robustness vs. reasoning: Xu et al. [1] focus on the robustness of NLP models in handling numerical data, which is a very important constitution of NLP and numeracy. On the other hand, our probing study pivots toward models' reasoning capabilities when dealing with numerical and tabular data. This means we delve deeper into understanding how well models can make sense of numerical relations in a structured format rather than just testing their resilience.
>
> *Comparing reasoning types:*
> - [1] gives interesting insights on models reasoning on two levels: a) number detection and extraction; b) semantic parsing of numbers
> - a) is comparable to our reasoning types "numeration". Similar to us, they evaluate number mapping between numerals and number tokens. They also include float numbers while studying number detection. We take a similar approach for reasoning type "heterogeneous numbers" but additionally include date, fraction, percentage, and scientific notation as heterogeneous number types in addition to floats. These number representations are commonly found in various real-world documents.
> - b) is similar to our reasoning type "arithmetic." However, while this work gives interesting insights into number parsing and using operands for calculations based on parsed numbers, our probes give a broader number of reasoning types and related challenges. For example, understanding the value of numbers by means of comparison between them, sorting numbers to understand their relation to each other, and so on.
> - Additionally, our work includes further reasoning types, which we motivate and discuss in detail in our paper.
>
> **Regarding probing additional models**
>
> Our goal on the model level was to cover a broad range of models. Hence, we included at least two models from three different categories (see Sec 5.1 for more details):
>
> 1. Numerical LMs: LUNA and PASTA (both are tabular reasoning models focusing on numerical data) NT5
> 2. LMs for tabular reasoning: TAPAS and DeBERTa with a table linearization approach
> 3. LLMs: FlanT5, GPT3.5 (to include both an open source model, i.e., FlanT5, and a closed model with high performance on various tasks, i.e., GPT3.5)
>
> The author further asked to include TAPEX [2]. While we considered doing so in the beginning, we selected TAPAS over TAPEX due to the following reasons (which we will include and clarify in the final paper):
> - TAPEX is pretrained by learning a neural SQL executor over database-type tables. However, our probes are not restricted to these kinds of tables but also contain InfoTabs infoboxes.
> - Moreover, we selected the TAPAS version from Eisenschlos et al. (2020) [3], which has a comparable performance to TAPEX.
> - To allow comparison to TAPEX, we have started probing TAPEX and will include probing results for the datasets other than InfoTabs to the final paper.
>
> For the final paper, we further plan to extend our three model categories with one additional model per category. Concretely, TAPEX and ReasTAP [4], which we have started probing over the last few days.
>
> **Typos, etc.**
>
> We thank the reviewer for pointing this out. We will fix all points mentioned in the final paper version.
>
> **References**
>
> [1] Jialiang Xu, Mengyu Zhou, Xinyi He, Shi Han, and Dongmei Zhang. 2022. Towards robust numerical question answering: Diagnosing numerical capabilities of NLP systems. EMNLP 2022.
>
> [2] Liu Q, Chen B, Guo J, et al. TAPEX: Table pre-training via learning a neural SQL executor.
>
> [3] Eisenschlos, Julian, Syrine Krichene, and Thomas Mueller. "Understanding tables with intermediate pre-training." Findings of the Association for Computational Linguistics: EMNLP 2020. 2020.
>
> [4] Zhao, Yilun, et al. "ReasTAP: Injecting Table Reasoning Skills During Pre-training via Synthetic Reasoning Examples." Proceedings of the 2022 Conference on Empirical Methods in Natural Language Processing. 2022.

---

### Meta-Review · Area_Chair_7DSb · 2023-09-19

**Recommendation:** 3

**Metareview:**

The reviewers found that the topic (of categorizing and analyzing numerical reasoning abilities of models) to be important and the empirical study extensive and systematic. There were some questions raised regarding the need for yet another taxonomy of numerical problems. The authors, in their response, clarified in detail important differences from prior work in the area. This would be very helpful to include in the revised version of the paper (as the authors indicated they would).

While it would be even better to expand the set of models used further to other table-specific models, I think the current coverage is broad enough to provide meaningful value to the readers.

The findings seem important and valuable enough to share with the community. At the same time, the paper lacks broad excitement, in part due to the existence of some prior efforts on analyzing numerical skills and in part due to the nature of this work being analysis-oriented rather than proposing a novel methodology that pushes the state of the art forward.

---

### Decision · Program_Chairs · 2023-10-07

**Decision:**

Accept-Findings

**Comment:**

The reviewers found that the topic (of categorizing and analyzing numerical reasoning abilities of models) to be important and the empirical study extensive and systematic. There were some questions raised regarding the need for yet another taxonomy of numerical problems. The authors, in their response, clarified in detail important differences from prior work in the area. This would be very helpful to include in the revised version of the paper (as the authors indicated they would).

While it would be even better to expand the set of models used further to other table-specific models, I think the current coverage is broad enough to provide meaningful value to the readers.

The findings seem important and valuable enough to share with the community. At the same time, the paper lacks broad excitement, in part due to the existence of some prior efforts on analyzing numerical skills and in part due to the nature of this work being analysis-oriented rather than proposing a novel methodology that pushes the state of the art forward.